# Recent Advances and Progress on Melanin: From Source to Application

**DOI:** 10.3390/ijms24054360

**Published:** 2023-02-22

**Authors:** Lili Guo, Wenya Li, Zhiyang Gu, Litong Wang, Lan Guo, Saibo Ma, Cuiyao Li, Jishang Sun, Baoqin Han, Jing Chang

**Affiliations:** 1College of Marine Life Science, Ocean University of China, Qingdao 266003, China; 2Laboratory for Marine Drugs and Bioproducts, Pilot National Laboratory for Marine Science and Technology, Qingdao 266235, China

**Keywords:** melanin, structure, synthesis, decomposition, bioactivity, application

## Abstract

Melanin is a biological pigment formed by indoles and phenolic compounds. It is widely found in living organisms and has a variety of unique properties. Due to its diverse characteristics and good biocompatibility, melanin has become the focus in the fields of biomedicine, agriculture, the food industry, etc. However, due to the wide range of melanin sources, complex polymerization properties, and low solubility of specific solvents, the specific macromolecular structure and polymerization mechanism of melanin remain unclear, which significantly limits the further study and application of melanin. Its synthesis and degradation pathways are also controversial. In addition, new properties and applications of melanin are constantly being discovered. In this review, we focus on the recent advances in the research of melanin in all aspects. Firstly, the classification, source, and degradation of melanin are summarized. Secondly, a detailed description of the structure, characterization, and properties of melanin is followed. The novel biological activity of melanin and its application is described at the end.

## 1. Introduction

Melanin originates from the Greek word “melanos”, which means black or very dark, reflecting the characteristics of melanin appearance; this term was first applied by the Swedish chemist Berzelius in 1840 to call a dark pigment extracted from eye membranes [1]. Melanin has a very long history as an ancient pigment that has been produced naturally since the beginning of life on Earth. Melanin can be found in well-preserved dinosaur fossils, prehistoric bird feathers, plants, marine cephalopods, bacteria, and fungi [2]. As early as 1840, “melanin” was used to refer to the black pigment in animals. Currently, melanin is commonly used to refer to a range of bio-pigments associated with most organisms, broadly defined as “heterogeneous polymers” formed by the polymerization of intermediate phenols and quinones from the oxidation of phenols or indole compounds [3].

Melanin can be found widely in nature, such as in the skin and hair of mammals, the ink of cephalopods, plants, and different types of bacteria and fungi [4]. It has a wide variety of functions in the biosystem. In animals, melanin is produced by melanocytes present in the epidermis and hair follicles and plays a role in sun protection and camouflage [5]. In plants, melanin acts as a reinforcer of cell walls and cuticles, increasing the resistance of plants to microbial and viral infections [6]. In addition, melanin protects microorganisms from environmental stresses, such as high sunlight exposure, low temperature, low water content, starvation, elevated reactive oxygen species, and increased radioactivity [7]. There are mainly three types of melanin, which are named eumelanin, pheomelanin, and allomelanin, respectively. Eumelanin is found in animals, microorganisms, and some fungi. It is derived from tyrosine and is black or brown in color [8]. Pheomelanin is endemic to higher animals, mammals, or birds. It is also a tyrosine derivative and is red or yellow in color. Pheomelanin consists of sulfur-containing monomer units, mainly benzothiazine, and benzothiazole, rather than the indole units in eumelanin [9]. Nitrogen-deficient plant and fungal melanin is often called allomelanin. Their precursors are different. Fungal melanin can be formed from γ-glutaminyl-3,4-dihydroxybezene, catechol, and 1, 8-dihydroxy naphthalene, while catechol, caffeic acid, chlorogenic acid, protocatechuic acid, and gallic acid have been considered as possible precursors in plants [3,10].

Melanin is a polymer formed by 5,6-dihydroxyindole (DHI) and its 5,6-dihydroxyindole 2-carboxylic acid (DHICA). However, its macromolecular structure is not very clear. Recent studies have demonstrated that the macromolecular structure of melanin has at least four layers and three particle types, which belong to supramolecular structures [11]. Moreover, the proportion of DHI and DHICA in the oligomer plays a very important role in the subsequent polymerization process. When the DHI ratio is large enough, the oligomers will have a π-stacking effect. Conversely, random bundling is performed due to its nonplanar structure [12]. Depending on the ratio and coupling site of DHI and DHICA, it can form large heteropolymers or laminated depressors. However, both structural forms of melanin have the same physiochemical properties [13]. Melanin is amorphous and predominantly dark brown to black, with red and yellow sometimes being observed [14]. Melanin is known to be resistant to concentrated acids, light, and reducing agents but is soluble in bases and phenols [15]. Additionally, it is susceptible to the oxidation that accompanies the bleaching processes. Moreover, it is extremely heat-resistant, even up to 600 °C [16,17]. 

There are many studies on the source and synthesis pathways of melanin, but the degradation of melanin has been rarely studied; therefore, the mechanism of melanin degradation remains unclear. Currently, lysosomal enzymes are involved in the degradation of melanin in keratinocytes. It has been found that isolated melanosomes can express lysosomal enzymes, such as acid phosphatase and cathepsins [18]. The expression of lysosomal hydrolases varies in the upper layers of the basal epidermis, which leads to different levels of melanin in keratinocytes of different tissues [19]. On the other side, considering the structure, the degradation of melanin may involve a redox mechanism similar to the polycyclic hydrocarbon conversion [20]. Melanin can be oxidized by oxidants such as potassium permanganate or hydrogen peroxide, with a slight bleaching phenomenon [21].

The indole structure or phenol group in the melanin structural unit confers many biological activities to melanin beyond its physical and chemical properties. Previous research has focused on free radical scavenging, chelating metal ions, and antioxidant and antibacterial activities of melanin and also described its application in industry, agriculture, and the food industry [13,22,23,24,25,26,27]. The anti-tumor [16,28], immunomodulatory [29], radiation protection [30,31,32], photothermal properties [33,34], and other activities of melanin have gradually attracted attention in recent years [35,36,37], and applied studies have gradually focused on biomedical aspects, such as magnetic resonance imaging (MRI) contrast agents [38,39], photothermal agents [40], and medical anti-tumor materials [41].

As discussed, many new results have been achieved in recent years in the studies and applications of melanin. Therefore, this review aims to provide a comprehensive summary of the current research progress on melanin. The review will begin with a summary of the classification, sources, and degradation of melanin. Then, the structure, characteristics, and physicochemical properties of melanin will be described in detail. The biological activities of melanin and its applications are described at the end of the review. Moreover, the future prospects of melanin are also summarized.

## 2. Classification and Origin of Melanin

Melanin is a heterogeneous polymer of phenolic or indoles, which is amorphous and has a wide range of functions. The colors of melanin are mainly from dark brown to black; sometimes, reddish and yellowish are also observed [3,42]. The widely accepted classification of melanin into eumelanin, pheomelanin, and allomelanin is based on the chemical composition of the pigment monomeric subunit structure. Recently, it has also been reported that melanin is classified into five major categories, namely eumelanin, pheomelanin, allomelanin, neuromelanin, and pyomelanin, respectively (Table 1) [43]. 

### 2.1. Eumelanin

Eumelanin is a heterogeneous polymer composed of 5, 6-dihydroxyindole (DHI) and 5, 6-dihydroxyindole-2-carboxylic acid (DHICA), which belongs to the aggregation network of oligomers (Figure 1). DHI and DHICA are dopachromes obtained from the oxidation of tyrosine or levodopa, which are further rapidly oxidized to cyclodopa by the cyclization of dopaquinone. The relative ratio of DHI and DHICA depends on the activity of dopa tautomer isomerase and the availability of copper ions, with a higher proportion of DHICA in general [44,45,46,47]. The predominant color of eumelanin is from black brown, which is found in human hair and skin. It is also produced by some bacteria, fungi, and myxomycetes [48,49]. Some myxomycetes can also produce melanin, and as a separate taxon of protists, myxomycetes can become dormant forms of plasmodia under adverse environmental conditions. Myxomycetes are able to produce melanin in spores, and melanin content is greater in dark spores than in bright spores [50]. The plant plasmodia P. nudum also produces melanin only when the organism is exposed to white light [51]. The particle size of natural eumelanin is not uniform. For example, natural eumelanin of human skin, eyes, and hair is composed of homogeneous particles 200 nm, and all have a spherical structure. It has also been found that the substructure of melanosomes extracted from bovine RPE cells is about 20 nm, which is similar to the substructure of squid melanin particles [52,53,54,55]. Therefore, there is a hypothesis that the 200 nm spherical structure of natural melanin is composed of an aggregation of smaller particle species. As for the aggregation mechanism of eumelanin, there is an updated version of the hierarchical accumulation consisting of four levels and three steps (Figure 2) [56]. Each aggregation step increases the particle size by one order of magnitude: The oligomeric sheets form proto-particles (10^−9^ m), which then arrange in an onion-like structure and densify into the smaller, spherical A-particles (10^−8^ m) and finally undergo aggregation to larger, anti-spherical B-particles (10^−7^ m) [56].

Eumelanin is found in many animals and has various effects. Bird feathers, for example, are a mixture of eumelanin and pheomelanin. Their levels vary with the species of bird. Interestingly, the concentrations of eumelanin and pheomelanin in female feathers are positively correlated with male feathers [58,59]. Eumelanin is also found in marine cephalopods, such as cuttlefish, whose ink contains a special kind of melanin called squid ink melanin. It is formed by irregular isomers of indole and carboxypyrrole units with some related proteins [3]. During the production of melanin, hydrogen peroxide, and free radicals are produced during its polymerization. The generated free radicals lead to the appearance of melanin units (such as carboxypyrrole) as a result of the rupture of the hexagon ring of the indole unit [60].

The carboxyl group in cuttlefish melanin has special metal chelating properties, which gives it superior ion exchange capacity beyond other types of melanin. Cuttlefish melanin also has the distinct advantage of being easy to extract in large quantities. This also makes it an excellent model for studying the structure and properties of eumelanin [3]. In addition, eumelanin has been found in reptiles and amphibians, where melanin has a role in changing color to suit the environment and where MCH (melanin concentration hormone) is active, leading to rapid and reversible transfer to certain skin areas [61]. There is also a special type of eumelanin, which is unique to insects. An insect’s exoskeleton has a sclerotic process that occurs simultaneously with melanization. The main precursor to this process is N-acetyl dopamine, which is derived from the amino acid L-tyrosine, which contributes to the specificity of this melanin [62]. 

### 2.2. Pheomelanin

Pheomelanin is derived from the common precursor dopaquinone, similar to eumelanin. However, L-dopa integrates cysteine to form dopa cysteine (CD isomer), which in turn is oxidized, cycled, and rearranged to form 1, 4-benzothiazine. Pheomelanin is polymerized from the final 1, 4-benzothiazine [42]. In the formation of the pheomelanin, L-cysteine or glutathione is added to multiple positions in the dopaquinone ring, but the predominant isomer produced is 5-S-cysteyl-dopa or 5-S-glutathione dopa [3,63]. Thus, the main structures of pheomelanin are benzothiazine and benzothiazole (Figure 3), which are oligomers of sulfur elements rather than indole units [64,65]. Benzothiazine- or benzothiazole-like pigment is mainly yellowish or reddish and is present in high levels in poultry feathers, human red hair, insects, and amphibians, as well as in reptiles, but in very rare amounts [66,67,68,69,70,71]. In addition, the presence of brown melanin in fungi has been reported, mainly based on the yellowish color of the extracted melanin and the presence of sulfur units detected by chemical analysis. However, according to the detection of melanin, most of the structure is indole, indicating that it may be eumelanin rather than brown melanin. Therefore, the existence of melanin in fungi is currently controversial [3,72,73,74].

### 2.3. Neuromelanin

As a mixture of eumelanin and pheomelanin, neuromelanin is the only melanin pigment that is not formed in melanocytes but in catecholaminergic neurons of the substantia nigra [75]. It has been shown that neuromelanin contains benzothiazine and indole units, with cysteine in its pheomelanin core and eumelanin on its surface [76]. Neuromelanin is a dark insoluble pigment produced in different areas of catecholaminergic neurons in the brain. Nigrasomes should contain pheomelanin in the core and eumelanin on the surface [77].

### 2.4. Allomelanin

Allomelanins are heterogeneous pigments with a nitrogen-free heterogeneous polymer class found in many fungi and plants. They come from many sources, including dihydrofolic acid, hyperic acid, catechol, etc. [78]. Allomelanin is a group of heterogeneous polymers emerging from the oxidation or polymerization of di-DHN (1,8-dihydroxy naphthalene) or THN (1,3,6,8-tetrahydroxy naphthalene) to produce DHN-melanin or allomelanin in various colors [79].

### 2.5. Pyomelanin

Unlike eumelanin, pheomelanin does not have its biosynthetic pathway but is associated with the activation of the L-tyrosine/L-phenylalanine degradation pathway. It is a water-soluble pigment that is produced during the accumulation and polymerization of homogentisic acid as a by-product of the tyrosine catabolic pathway [80]. In the tyrosine catabolic pathway, the deamination of tyrosine generates p-hydroxyphenylpyruvate, which is oxidized to homogentisic acid by further oxidation and decarboxylation. The final formation of pheomelanin is achieved through cyclic rearrangement and successive polymerization of homogentisic acid [81].

### 2.6. Other Types of Melanin

In addition to the aforementioned melanins, other less common melanins are occasionally produced. Trichochrome pigments (formerly known as trichosiderins) are formed in the same metabolic pathways as eumelanin and pheomelanin and have a lower molecular weight than the original melanin [82]. In addition, there are fungal melanins with specific molecular characteristics, most of which are synthesized by ascomycetes through the polyketone compound pathway, using 1,8-dihydroxynaphthalene (1,8-DHN) as a precursor [3].

## 3. Synthesis and Decomposition of Melanin in Nature

### 3.1. Synthesis of Melanin in Humans and Animals

In humans and animals, melanin is produced in specialized cells called melanocytes, mainly at the epidermal-dermal junction, and then to the surrounding keratinocytes, which are the most abundant cells in the epidermis. Melanocytes look like dendritic cells and contain specialized lysosomal lineage organelles called melanosomes, which synthesize and store melanin. Melanosomes are transferred from melanocytes to neighboring keratinocytes by elongated dendrites [83]. The synthesis of melanin in humans follows the Raper-Mason pathway, in which L-tyrosine is synthesized in an environment with or without cysteine through a series of enzymatic reactions catalyzed by tyrosinase, resulting in the production of true melanin or brown melanin, a pathway first proposed by Raper in the 1920s and later demonstrated by Mason using spectrophotometry [84,85]. The biochemical pathway of melanin synthesis begins with the amino acid L-tyrosine, and the key enzyme is tyrosinase. Tyrosinase catalyzes two sequential reactions; one is the hydroxylation of L-tyrosine to L-dopa. L-tyrosine, which is delivered by the gastrointestinal tract (GI) or produced by L-phenylalanine hydroxylase (PH) -mediated hydroxylation of L-phenylalanine, can be hydroxylated to L-dopa by tyrosine hydroxylase (TPH) or tyrosinase (Tyr), depending on the cell type [86]. Tyrosine hydroxylation catalyzed by tyrosinase was first proposed by Aaron Bunsen Lerner [87].

The other is the oxidation of tyrosine by tyrosinase to produce dopaquinone. In the absence of cysteine, dopamine quinone undergoes an intramolecular addition of an amino group to produces cyclic dopa (white dopachrome). Redox exchange between white dopachrome and dopaquinone produces dopachrome and dopa. Dopa is the noncyclic reduction product of dopaquinone, while dopachrome is the cyclic oxidation product of cyclodopa. Dopa is recruited into the pathway by tyrosinase, and dopachrome continues to form eu-/pheo-melanin [88]. Dopachrome slowly and spontaneously decarboxylates to 5, 6-dihydroxyindole-2-carboxylic acid (DHI), and dopa pigment tautomerase (Tyrp2/DCT) can catalyze the formation of 5, 6-dihydroxyindole-2-carboxylic acid (DHICA) from dopachrome (Figure 4). DHI and DHICA are indole-o-diphenols that are readily oxidized to their corresponding o-quinones by tyrosinase or tyrosinase-associated protein 1 (Tyrp1). These species eventually aggregate to form eumelanin [89]. DHI and DHICA can be oxidized to their respective o-quinones by oxygen and ROS produced in the previous reaction without any enzymes. The post-dopa oxidase steps of melanogenesis were first described by John Pawelek [90].

In any case, eumelanin is a polymer with a mixture of decarboxylated (DHI) and carboxylated (DHICA) 5, 6-oxidized indole units (5, 6-dihydroxyl, semiquinone, and 5, 6-quinone units) of varying degrees of oxidation [91]. On the other hand, dopa quinones can bind to thiol-containing compounds such as free L-cysteine or glutathione, thereby branching into the production pathway of sulfur-containing that produce pheomelanin. As a result, human skin contains two types of melanin, eumelanin, and pheomelanin. Eumelanin is dark and varies from black to brown depending on the DHI/DHICA ratio, whereas pheomelanin is a red or yellow pigment and is found mainly in superficial, blonde, or red hair [92,93]. On the other hand, in the presence of cysteine, dopaquinone reacts rapidly with cysteine to produce 5-S-cysteine dopa and, to a lesser extent, 2-S-cysteine dopa. Cysteinyl dopa is then oxidized to generate the benzothiazine intermediate and, finally, pheomelanin [94].

L-dihydroxyphenylalanine (L-dopa) is the product of enzymatic hydroxylation of L-tyrosine, which is decarboxylated to dopamine and further processed to norepinephrine or epinephrine in neurons or chromaffin cells [95]. The non-essential aromatic amino acid, L-tyrosine, in addition to protein synthesis, also acts as a precursor for melanin, catecholamines, tyramine/octopus, and thyroid hormones [96]. L-dopa and L-tyrosine have been recognized not only as continuous substrates and intermediates of melanogenesis but also as positive regulators essential for melanogenesis. In Bomirski hamster melanoma cells, L-tyrosine supplementation not only increased melanin formation but also enhanced tyrosine hydroxylase activity and tyrosinase oxidation activity to dopa. L-dopa exerts a dose-dependent stimulatory effect on tyrosinase activity and melanin deposition by expanding the melanocortin (MSH) receptor system [97,98,99]. L-dopa itself does not affect the melanogenic pathway in melanoma cells, but at micromolar or lower concentrations, it stimulates cell proliferation [100].

L-tyrosine and L-dopa can synergize with UVR to increase MSH receptor expression activity and promote melanin synthesis. L-tyrosine, as well as phosphorylated isoforms of L-dopa, can stimulate cell surface MSH receptor expression, increase the level of MSH stimulation of tyrosinase, and reduces positive cooperativity among cell surface MSH receptors [101,102,103]. Similarly, UVB upregulates a-MSH receptor (MC1R) expression and activity [104], POMC expression and production of POMC peptides, including a-MSH, b-endorphin, and ACTH [105,106], potentially regulating mammalian skin pigmentation, protecting skin from UV-induced damage and modulating skin immune responses [107,108,109].

During melanogenesis, some metal cations may have a significant effect on the rate of formation and the final structure of the synthesized melanin [110]. The degree of influence varies from one metal ion to another, and these differences depend on the nature of the ions, the pathway steps followed, and the presence of dopachrome tautomerases in the reaction mixture. Regarding the first step of the pathway, tyrosine hydroxylation, Co (II), and Ni (II) enhanced this activity, but Zn (II) inhibited it. In the second part of the pathway, the presence of metal ions also increases the rate of conversion of dopachrome to indole, inhibits decarboxylation, and enhances the binding of carboxylated precursors to the final pigment, with a less pronounced effect of Zn (II). Ni (II) and Co (II) stimulated melanin formation from L-tyrosine and L-dopa catalyzed by tyrosinase, but the inhibition of tyrosine hydroxylation by Zn (II) inhibits melanin formation. The activation induced by Ni (II) and Co (II) may originate from the direct reduction of copper at the tyrosinase active site. The differential inhibition of this reaction by Zn (II) does not exclude the inactivating effect of this ion on melanoma tyrosinase [111]. In the second phase, the combination of cations (except Zn (II)) with dopamine tautomerases inhibits decarboxylation, and the combined effect of cations and enzymes always inhibits decarboxylation more than each individually [110,112].

In addition to metal ions, hormones also play an important role in the process of melanogenesis. Locally produced melanocortin (MSH) and adrenocorticotropic hormone (ACTH) can regulate melanogenesis by paracrine, autogenous, or endocrine mechanisms [113]. In humans, MSH or ACTH stimulates hyperpigmentation of sun-exposed skin [114]. Chronic synthetic ACTH leads to skin atrophy, hyperpigmentation, and hirsutism (see references [113,115], and high serum MSH concentrations are also associated with skin hyperpigmentation. Structurally, the MSH peptide and ACTH share the same amino acid sequence -Tyr-X-Met-X-His-Phe-Arg-Trp- containing the tetrapeptide -His-Phe-Arg-Trp- essential for melanic activity. Polymorphisms in MC1 receptors are associated with skin and hair hyperpigmentation. Mutations that reduce MC1 receptor activity leads to lighter skin pigmentation and redness; the external environment, such as ultraviolet radiation or the high level of MSH or ACTH in the plasma, can combine with the over-expressed MC1 receptor in melanocytes, leading to skin or hair hyperpigmentation [108,116,117].

It should be noted that melanogenesis is a metabolic pathway specific to normal and malignant melanocytes that can influence the behavior of their surrounding cells or environment. Melanosomes can alter cellular energy production metabolism by converting oxidative catabolism to anaerobic glycolysis, altering intracellular NAD/NADH and NADP/NADPH ratios, and stimulating the pentose phosphate pathway [118,119,120]. Melanogenesis, however, produces an oxidative environment in which some of its intermediates, quinones, and semiquinones, are directly toxic but also mutagenic. Therefore, the mutational environment associated with melanogenesis may contribute to genetic instability. In addition, since melanogenic intermediates can suppress the activity of immune cells, an immunosuppressive environment may surround the tumor. Finally, the final product, melanin, can scavenger free radicals and reactive oxygen species (ROS), resulting in a relatively hypoxic environment due to increased oxygen consumption [121,122].

### 3.2. Synthesis of Melanin in Plants

Melanin has been shown to be widespread in the plant kingdom, and the biochemistry and molecular genetics of melanin formation in plants is less well-studied compared to animals and microorganisms. In addition to the complex polymeric nature of pigments, one reason is that plant melanin accumulates in the hard seed envelope, where other compounds with similar colors may be present, such as proanthocyanidins [10]. Some plants have been shown to accumulate melanin in their seeds by physicochemical methods, such as Castanea mollissima, Avena sativa, Helianthus annuus, Citrullus lanatus, Hordeum vulgare, Fagopyrum esculentum, Vitis vinifera, and Ipomoea purpurea [123]. Melanin synthesis in plants and polyphenol oxidase (PPO) in the damaged tissues of enzymatic browning reaction, polyphenol oxidase (PPO) can be used in the presence of low oxygen to phenolic copper redox enzyme family due to aging, damage, and the interaction of pests and pathogens and in the process of post-harvest processing and storage, processing. Loss of compartment integrity in cells results in the release of PPO from the plastids in which they reside into the cytoplasm. PPO contacts vacuolar phenolic substrates and forms highly reactive o-quinones. O-quinone subsequently polymerizes nonenzymatically or interacts with other compounds, such as thiols, amino acids, and peptides, then it forms colored products. O-quinone can also interact slowly with water to form triphenols or reduce to primitive phenols [124,125,126].

### 3.3. Synthesis of Melanin in Microorganisms

Several bacteria have been reported to produce different melanin genomes by specialized pathways or by exploiting enzymatic imbalances in modified metabolic channels [78]. The regulation of the melanin synthesis mechanism in bacteria includes transcriptional regulation and metabolic regulation. Melanin biosynthesis is present in Gram-positive, and Gram-negative bacteria, such as Streptomyces grays, Bacillus licheniformis, and Rastonia solanacearum [127,128,129]. Fungi have been reported to have all types of melanin. Melanin in fungi is considered to be a secondary metabolite that can be synthesized from endogenous substrates via 1,8-dihydroxynaphthalene (DHN) intermediates or L-3, 4-dihydroxyphenylalanine (L-dopa) [130]. Although melanin is not essential for the growth and development of fungi, it can perform a wide range of biological work. For example, it is essential for host invasion of plant pathogens. Melanin in the cell walls provides mechanical strength to the appressorium and aids in tissue penetration [131].

Typically, most microbial melanin is formed by the conversion of tyrosine (DOPA pathway) or malonyl-CoA (DHN pathway), facilitated by different enzyme genomes. The dopa pathway is very similar to mammalian melanin synthesis. The melanin precursor tyrosine is converted to levodopa, which is then converted to dopaquinone by tyrosinase and laccase. Dopaquinone has high activity and can spontaneously oxidize and self-polymerize to form melanin. Synthesis of melanin through the dopa pathway is called dopa melanin or eumelanin [27]. The precursor of the DHN pathway is propylene glycol—coenzyme a. Polyketo compound synthase catalyzes the continuous decarboxylation of five malonyl-CoA molecules to produce 1,3,6, 8-tetrahydroxy naphthalene (THN), which through a series of reduction and dehydration reactions generates 1, 8-dihydroxy naphthalene (DHN), and finally polymerizes into DHN melanin. Notably, both pathways can be found in bacteria and fungi. Most bacteria and basidiomycete fungi synthesize melanin through the dopa pathway. However, ascomycetes and some bacteria, including non-microscopic fungi, such as truffles, use the DHN pathway to produce melanin [17,79,130].

### 3.4. Decomposition of Melanin

The metabolic process of melanin mainly includes the synthesis, transport, and degradation of pigment. Currently, most of the research is limited to the synthesis and transport of melanin, especially in the synthesis; however, studies of melanin degradation are rarely involved, and the mechanism of its occurrence is still unclear [132,133].

Melanocytes are cells found in the skin tissue that produce melanin. Melanosomes are large organelles responsible for the synthesis, storage, and transport of melanin. Mature melanosomes are transferred from melanocytes to the nucleus of keratin-forming cells to form hyperpigmentation [134]. Melanocytes use long dendrites to contact many keratinocytes (one epidermal melanocyte provides pigment to more than 40 keratinocytes) and transfer melanosomes from the site of formation in the center of the cell to the main site of transfer at the tip of the dendrite. Melanocytes accomplish this task by combining the long-range, bidirectional microtubule-dependent transport of melanosomes along the length of the dendrite with the actin-based motor protein myosin capture and local movement of organelles in the actin-rich dendritic tips. Finally, melanosomes accumulated at the tip of dendrites by this cooperative trapping mechanism are transferred out of melanocytes and into keratinocytes. This helps distribute pigment in the hair and skin [135,136]. In addition to being associated with melanogenesis, melanocytes also act as sensing and regulatory cells in the human epidermis. For example, melanocytes can act as intra-epithelial stress sensors, alter KC functions by the transfer of melanosomes, regulate immunity, serve as neuroendocrine cells similar to APUD cells for amine-precursor-uptake and decarboxylation, amplify and convert signals collected from adjacent cells into chemical messages to maintain cellular homeostasis [137].

Lysosomal enzymes are involved in the degradation of melanin in keratogens. Lysosomes are cytoplasmic vesicles surrounded by a membrane monolayer, widely distributed in all mammalian cells except erythrocytes, and play an important role in cell proteolysis. Currently, lysosomes have been shown to contain more than 60 kinds of hydrolases, which can degrade and digest proteins, nucleic acids, and polysaccharides [138].

Lysosomal enzymes are associated with melanosome degradation. Melanosomes are advanced subcellular organelles composed of many components. Melanin is its most distinctive part and major part. Melanosome disintegration is a gradual process. The edges of slightly disintegrated pigment granules appear as folds, which are the melanin fragments nearby. With further disintergration melanosomes are disrupted, and their density decreases. At late stages, the underlying stromal structure may become visible, with melanin particles mixing with the matrix material [139]. Melanosomes lose their integrity in keratinocytes and their lysosomes and are converted into ill-defined melanosome dust, which represents one of the self-assembling units of eumelanin. The products of melanosome disintegration remain in the lysosomal compartment until they are lost by epidermal desquamation [140,141,142,143,144]. All these studies suggest that lysosomal enzymes in keratinocytes play an important role in the degradation of melanin. Other studies have found that there is differential expression of lysosomal hydrolases (B3GTL, Cath B, Cath L2, Cathy, ACPP, β-GLU) in the upper basal epidermis, which leads to different melanin content in keratinocytes from different tissues. The efficiency of melanin degradation by keratinocytes in white skin is higher than that in black skin, and the degradation of melanin shows a time-dependent manner [145,146].

After treating radiolabeled melanosomes with lysosomal hydrolases, Japanese scientists noted that lysosomes isolated from hepatocytes did not appear to degrade melanin by themselves but that lysosomal enzymes hydrolyze the core protein fraction in melanosomes. The melanosomes are mainly composed of an outer membrane and an inner membrane matrix, and the main components are probably proteins on which the melanogenesis of polymerized indole 5, 6-quinones, and other tyrosine intermediates take place [147].

The basic molecular unit of eumelanin is thought to be a small planar oligomer composed of several 5, 6-dihydroxyindole-2-and 5, 6-dihydroxyindole-2-carboxylic acid units, which are further assembled into a stable high-order parallel layer structure by P-stacking and lateral interactions [148]. Therefore, considering its structure, a redox mechanism similar to polycyclic hydrocarbon conversion may be involved in the degradation of melanin [149]. This idea is further supported by several reports. Melanin can be degraded by oxidizing agents such as potassium, permanganate [150], and hydrogen peroxide [151], and the degradation process is accompanied by mild bleaching [21] and a strong fluorescence phenomenon [152,153,154,155]. In addition, potassium permanganate and hydrogen peroxide oxidation are also the basic methods used to quantify melanin [156].

## 4. Macromolecular Structure and Physicochemical Properties of Melanin

### 4.1. Macromolecular Structure of Melanin

The structure of melanin is closely related to its properties and functions. Therefore, it is very important to elucidate the macromolecular structure of melanin to get a better understanding of its properties. It is generally believed that melanin is a heteropolymer formed by DHI and DHICA in a certain ratio (Figure 4). The ratio of these two components depends on the type and synthesis pathway of eumelanin, which ultimately leads to changes in the properties, such as appearance, oxidation resistance, metal chelation performance, etc. [157]. The content of DHICA of eumelanin prepared by the enzymatic method is only 10%, while natural eumelanin could reach up to 50% [158].

Eumelanin synthesis involves the enzymatic oxidation of tyrosine or dopa to give dopachrome, which undergoes isomerization to 5,6-dihydroxyindole-2-carboxylic acid (DHICA) with tyrosinase-related protein (Tyrp2). In the absence of Tyrp2, dopachrome spontaneously decarboxylated into 5,6-dihydroxyindole (DHI) [159]. DHICA melanin exhibits more potent hydroxyl radical-scavenging properties than DHI. Moreover, DHI melanin is composed mainly of planar oligomeric scaffolds, while DHICA is composed of distorted linear oligomeric structures characterized by resistive isomerism caused by slow rotation around interunit bonds [160].

Polydopamine (PDA) is a supramolecular aggregate of monomers containing three main types of structural units, namely, uncyclized amine-containing units, cyclized eumelanin-type indole, and pyrrole carboxylic acid units, respectively [161]. Therefore, PDA consisted of a mixture of oligomers in which indole units and open-chain dopamine units with different levels of unsaturation produced charge-transfer interactions between o-quinone and catechol units [162].

Basic eumelanin particles are approximately 15 A in size, consisting of four to five layered stacks with a stack spacing of approximately 3.45 A, and are composed of four to eight molecules composed of 5, 6-dihydroxyindole monomers in a plane [163]. However, graphite-like nanostructures, about 15 nm in size and consisting of dozens of stacked carbon layers, are found to exist in carbonized PDA nanoparticles by using high-resolution TEM and Raman spectroscopy [164].

### 4.2. Physicochemical Properties of Melanin

The solubility of melanin varies according to the source, purity, and polymerization state. Most melanin is virtually insoluble in distilled water and most organic and inorganic solvents and will precipitate under acidic conditions [165]. However, it is soluble in dimethyl sulfoxide (DMSO) and alkaline water (pH 10.0) [2]. The solubility of melanin highly depends on the pH of the solution. The decrease in the pH value of melanin solution will promote the formation of aggregation and precipitation [43], while the increase in the pH value will lead to the decomposition of particles into small particles with a poor degree of polymerization, which is called oligomers. The presence of intramolecular ionizable units and hydrophobic interactions is responsible for this behavior [166]. The solubility of melanin is also related to the ionization of carboxyl, phenol, and amine groups in the molecule, polyelectrolyte characteristics, and amino acid content [167].

Melanin has unique reaction properties that can be used for the initial detection and characterization of melanin. The bleaching nature of melanin is one of its characteristics. Pigment decolorization occurs in the presence of potassium permanganate, potassium dichromate, sodium hypochlorite, hydrogen peroxide, or other oxidizing agents, which is related to the degradation of the pigment [43]. The reaction mechanism with H_2_O_2_ is nucleophilic attack by OOH^−^, which induces a ring-opening reaction leading to the formation of quinone epoxides that bleach the melanin. The reactivity of melanin also includes the reaction with the AgNO_3_ solution. Gray precipitation will appear on the tube wall, which is caused by the reduction of AgNO_3_ [42].

## 5. Biological Activity of Melanin

Melanin contains phenolic hydroxyl, carboxyl, and amino active groups, which can absorb UV light, and is a natural endogenous functional substance. The biological activities of melanin are closely related to its composition and structure. Among the properties of melanin, the representative ones are free radical scavenging, antibacterial, chelating metal, anti-tumor, and radiation protection activities, which have gradually become the hot spot and trend of melanin activity research.

### 5.1. Activity of Free Radical Scavenging

Free radicals are highly reactive substances with strong oxidative effects. Excessive free radicals can damage proteins, nucleic acids, and cell membranes, leading to damage to normal cells and tissues in the human body [168]. In addition, excessive free radicals in the body are also closely associated with the onset, development, and aging of many diseases in the human body, which is the major cause of human diseases.

Back in 1998, There are scholars suggested that melanin could act as a natural scavenger of highly reactive nitrogen elements, and they found that the NO_2_ radical produced by the oxidation of nitrite by lactoperoxidase and H_2_O_2_ can react with the melanin 5, 6-dihydroxyindole subunit to oxidize it to the corresponding semiquinone radical [169]. Recently, there is a study also demonstrated that melanin can act as a natural scavenger of nitrous acid and some nitrous acid-derived species [170].

Melanin has perfect free radical scavenging activity. Zou et al. found that melanin has strong scavenging activity against DPPH radical, superoxide radical, and hydroxyl radical, with half inhibitory concentration of 0.18, 0.59, and 0.34mg/mL, respectively, and the scavenging ability was concentration dependent. With the increase in concentration, the scavenging effect was enhanced, which was significantly higher than that of antioxidants in the market [171]. By isolating and identifying melanin from Marine Actinobacter MA32, Manivasagan P studied the scavenging activity of melanin free radicals at different concentrations (0–3.5 mg/mL) and found that it had significant scavenging ability on DPPH free radicals, superoxide free radicals and nitric oxide. The IC50 values were 85, 71, and 68 mg/mL, respectively, and the effect was more significant than that of common antioxidants BHA and ascorbic acid [25]. Li et al. found that melanin components extracted and purified from Streptomyctus showed significant scavenging activity of superoxide and hydroxyl radicals, and the scavenging capacity increased with the increase of concentration, showing a strong dose-response relationship [172]. Gonçalves et al. extracted melanin from Aspergillus Niger and studied its antioxidant activity. It was found to be effective in inhibiting the oxidative capacity of HOCL and H_2_O_2_ and scavenging free radicals generated by the oxidation of HOCL and H_2_O_2_ in a concentration-dependent manner [173]. Peng et al. investigated the free radical scavenging rate of melanin extracted from Citrus aurantium seed entities and showed that DPPH achieved 63.04% free radical scavenging rate. When the concentration was 0.375 mg/mL, the scavenging rate of the superoxide anion reached 39.79%, but the antioxidant activity of hydroxyl radical was weak [174].

### 5.2. Antimicrobial Activity

Bacterial infections are often the main cause of surgical failures and wound infections. It is very important to find an antimicrobial agent with good biocompatibility, biosafety, and environmental friendliness. Rahmani Eliato et al. found that natural melanin extracted from horse mane could significantly inhibit the growth of Escherichia coli and Staphylococcus aureus after co-culture, and the growth inhibition rate of bacteria could reach 100% within 4 h (Figure 5) [175].

Li et al. demonstrated that melanin of auricularia had a significant inhibitory effect on bacterial quorum-sensing regulation behavior, and when melanin of auricularia was 80 μg/mL, the inhibition rates of biofilm formation against Escherichia enterica K-12, Pseudomonas aeruginosa PAO1, and Pseudomonas fluorescent P-3 were 71.3%, 61.7%, and 63.2%, respectively [23]. Xu et al. investigated the effect of intracellular melanin from Trichoderma YM30 on Gram-negative bacteria, *Escherichia coli*, Salmonella typhimurium, Vibrio parahaemolyticus, Gram-positive bacteria, Bacillus monocytogenes, Bacillus giant and Staphylococcus aureus. It was found that melanin disrupted the integrity of the cell membrane, increased the leakage of cell contents, increased the uptake of non-protein nitrogen, decreased the membrane potential, and had significant antibacterial activity [176]. Liu et al. verified the inhibitory effect of melanin extracted from mussel shell by HCL hydrolysis (H-melanin) and trypsin hydrolysis (T-melanin) on *Escherichia coli* and Staphylococcus aureus and found that the antibacterial effect of the latter was significantly stronger than that of the former. After 15min of NIR irradiation, the inhibitory rates of t-melanin and h-melanin against escherichia coli and staphylococcus aureus were 97.43% and 94.23%, 35.43%, and 29.10%, respectively (Figure 6) [177].

### 5.3. Activity of Radiation Resistance

UV radiation is harmful and excessive exposure to UV radiation can lead to various skin cancers and other adverse health effects. The skin is the organ most exposed to environmental UVR and associated sequelae [178]. Ultraviolet radiation (UVR) causes three main forms of skin cancer: basal cell carcinoma (BCC), squamous cell carcinoma (SCC), and cutaneous malignant melanoma (melanoma). Prolonged unprotected exposure to UVR weakens skin elasticity, leading to sagging cheeks, deepening facial wrinkles, and skin discoloration [179,180]. In addition, ultraviolet radiation is also very harmful to the eye. Both UV-A and UV-B can cause cataract formation, which in the long term can lead to visual impairment and temporary or permanent blindness [181]. Therefore, radiation protection is very important.

Radiation damage is one of the most common causes of harmful changes in the human body. Light absorption is the most prominent characteristic of melanin polymers, which is caused by highly conjugated structures and protect against light radiation in human. Li et al. proved that melanin (LIM) and its arginine-modified derivatives (ALIM) produced by Choriococcus granulosa YM156 had a certain protective effect on UV-B-induced injury in mice, and could reduce radiation-induced oxidative and inflammatory damage, indicating that LIM and ALM had radiation-protective effects on UVB radiation-induced injury in mice. In addition, LIM and ALIM inhibited the overexpression of proinflammatory cytokines, including interleukin (IL)-1α, IL-1β, IL-6, and tumor necrosis factor-α (TNF-α). Similarly, Ye et al. found that melanin secreted by Trichoderma granulosa could significantly improve the survival rate of *Escherichia coli*, *Staphylococcus aureus*, and *Saccharomyces cerevisiae* under ultraviolet irradiation through in vitro radiation resistance experiments. The radiation test in mice showed that it had strong activity against ultraviolet radiation [30]. Vijayan et al. proved through ROS and MTT experiments that melanin secreted by sponge-related bacteria had a certain protective effect on L929 cells induced by UV [182]. Kunwar et al. found that melanin could prevent and alleviate radiation damage to mice by improving the reduction of ERK phosphorylation in spleen tissue, hematopoietic injury, spleen index, total cell count, white blood cell count, and platelet count. At an effective dose of 50 mg/kg, melanin exhibited effective radiation protection [183].

### 5.4. Activity of Chelating Metal Ion

Heavy metals, including lead, mercury, cadmium, copper, arsenic, etc., accumulate in the human body and lead to chronic poisoning and various diseases. Heavy metals, distributed in the atmosphere, water, and soil, are difficult to be biodegraded. There are many active sites and functional groups in the structure of melanin, such as carboxyl, amine, hydroxyl, and carbonyl groups, which can bind to heavy metals and subsequently eliminate their toxicity [184].

Manirethan et al. investigated the adsorption capacity of melanin synthesized by marine bacteria on mercury, chromium, lead, and copper at different pH, time, initial concentration, and temperature conditions. The result showed that the maximum adsorption capacity of Hg, Cr, Pb, and Cu was 82.4 mg/g, 126.9 mg/g, 147.5 mg/g, and 167.8 mg/g, respectively (Table 2). The binding effect of heavy metals on the surface of melanin was confirmed by FT-IR spectroscopy and X-ray photoelectron spectroscopy [185].

Szpoganicz et al. found that three functional groups in the eumelanin structure play a key role in the process of binding to metal ions. Zn bound more easily to quinone-imine in weakly acidic media, and Zn and Cu bound only to catechol groups in strongly alkaline conditions [186].

Vinay et al. revealed that melanin could combine with transition metals such as Fe, Zn, Cu, Pb, Mn, and Ti in addition to heavy metals [187]. Lian et al. quantitatively studied the combination of magnesium, calcium, zinc, copper, and iron with cuttlefish black pigment. It was concluded from the experiments that Mg, Ca, and Zn are bound to carboxyl groups in melanin, Cu bound to hydroxyl (OH), and Fe (II) bound to OH or amine groups [188]. Yu et al. studied the adsorption of Cr6+ ions in wastewater by melanin secreted by Mildew pullulan. The optimal adsorption conditions were 30 °C, pH 3, and 2 h by orthogonal horizontal experiments. The optimal adsorption rate was 27% [189].

### 5.5. Activity of Tumor Inhibition

Melanin has shown good antitumor activity in many aspects. Firstly, melanin has light-absorbing activity and can protect against skin cancer caused by light effects such as ultraviolet radiation. El-Naggar et al. examined the antitumor activity of melanin produced by Streptomyces sp. against skin cancer cell lines in vitro using the MTT method and found that the inhibitory activity was very strong. Melanin at a concentration of 50 µg/mL could inhibit 70.9% of cellular activity with better compatibility than 5-fluorouracil [16]. Shi et al. investigated the anti-hepatocarcinogenic activity of Lachnum YM226 melanin and its arginine derivative (ALM) using the H22 liver cancer mouse model. The results showed that both LM and ALM could effectively inhibit tumor growth in H22 tumor-bearing mice. There were also improvements in body weight, liver, spleen, and thymus indices in the LM and ALM groups. In addition, both of them could also reduce alanine aminotransferase (ALT), aspartate aminotransferase (AST), alkaline phosphatase (ALP), creatinine (CRE), blood urea nitrogen (BUN), and uric acid (UA) levels. The antitumor effect of ALM was significantly better than that of LM at the same dose [28]. Al-Obeed et al. found that herbal melanin (HM) had inhibitory effects on colorectal cancer HT29 and mCRC SW620 cell lines. HM-induced apoptosis was associated with mitochondrial outer membrane permeability and cytochrome C release, inhibition of Bcl2 family proteins, and activation of caspase-3/-7. In addition, HM regulated the MAPK pathway by the activation of the JNK pathway and the inhibition of ERK phosphorylation. TLR4 receptor down-regulation enhanced HM-induced apoptosis, while TLR4 receptor blockade partially attenuated HM-inhibited ERK phosphorylation [190].

However, melanogenesis is not beneficial in suppressing all tumors. Brozyna et al. noted that melanin could use its own photo-radiation characteristics to enhance the sensitivity of advanced melanoma to ionizing radiation and weaken the efficacy of radiotherapy [122]. Melanogenesis can reduce the killing effect of cyclophosphamide and IL-2 activated peripheral blood lymphocytes on melanoma cells and make them resistant to chemotherapy drugs and lymphotoxicity [121]. Inhibition of melanin production by blocking the catalytic site of tyrosinase or chelating copper ions, sensitizing melanoma cells to the cytotoxic effect of cyclophosphamide, and enhancing the immunotoxicity activity of IL-2-activated lymphocytes are effective targets for the treatment of advanced melanoma [32,191,192].

### 5.6. Immune Regulatory Activity

Li et al. studied the modulatory effects of melanin and its derivatives extracted from the fermentation broth of Trichoderma on cyclophosphamide-induced immunodeficient mice. It was found that melanin and its derivatives could significantly improve the specific, non-specific, humoral, and cellular immune functions in mice [29]. Kunwar et al. confirmed the radiation protection effect of melanin on mice. In this process, melanin reduced the production of proinflammatory cytokines (IL-6 and TNF-α) and reduced oxidative stress in liver tissue, thus eliminating the immune imbalance that might be caused by radiation damage [183].

### 5.7. Photothermal Characteristics

Natural melanin or synthetic polydopamine nanomaterials have a good light absorption effect and can convert near-infrared light into heat energy, showing good photothermal conversion efficiency. In addition, this nanomaterial has good biocompatibility and degradability and does not cause long-term toxicity to the human body. Jiang et al. coated melanin nanoparticles with erythrocyte membrane and McF-7 membrane and injected them intravenously into tumor-bearing nude mice to accumulate melanin nanoparticles at the tumor site. After continuous irradiation with an 808 nm laser for 4 h, the surface temperature at the tumor site could reach 54 °C, which could effectively destroy the cancer cells. Applying the mixed membrane on the nanomaterials could reduce the loss of the immune system and blood circulation to the nanomaterials. When the membrane protein weight ratio of the two membranes was 1:1, it had a higher accumulation rate and photothermal conversion effect [33]. Zhou et al. developed a pegylated melanin-like nanoparticle modified by the dipeptide RGD and Beclin 1. It could achieve effective inhibition of tumor growth at temperatures around 43 °C. Beclin 1 could enhance the autophagy of tumor cells, and RGD could enhance the cellular uptake of melanin-like nanoparticles by tumor cells, to effectively inhibit tumors under the dual strategy of improving tumor autophagy and photothermal therapy (Figure 7) [34].

Li et al. designed a strategy of photothermal therapy based on natural melanin nanoparticles to induce immunogenic cell death in breast cancer. Melanin extracted from cuttlefish ink was coated with 4T1 cell membrane to achieve homologous adhesion of nanoparticles to tumor cells, which was then injected intravenously into mice. Compared with the control group, the levels of IL-6 and IL-12 were significantly increased after 12 h of NIR irradiation, and T cells in tumor cells were also significantly increased. The results indicated that the immune response was stimulated, and severe damage was caused to tumor tissue [193]. Kim et al. prepared a poloxamer thermosensitive gel containing melanin. After intratumoral injection, the temperature of the tumor tissue could reach 55 °C within 3 min of 808 nm near-infrared light irradiation. After 8 days of continuous laser irradiation, the tumor was completely inhibited without recurrence [40].

### 5.8. Other Biological Activities

In addition to the above activities, scientists have found that melanin also has liver injury relief, DNA protection, and anti-viral activities. Hou et al. found that black fungus melanin could protect mice from ethanol-induced liver injury. The liver index, serum alanine aminotransferase (ALT), aspartate aminotransferase (AST), γ-glutamyl transpeptidase (γ-GT), and hepatic malondialdehyde (MDA) levels were significantly decreased in melanin-treated mice. In addition, the levels of antioxidant enzymes such as alcohol dehydrogenase (ADH), catalase (CAT), and superoxide dismutase (SOD) were increased [37]. Rageh et al. found that irradiated mice showed decreased activity of antioxidant enzymes, increased the level of malondialdehyde, and increased DNA damage by 3–10 times. Treatment with melanin restored antioxidant enzyme activity and reduced malondialdehyde production, which protects cells from DNA damage and death [36]. Hamanaka et al. found that melanin could inhibit the synthesis of abnormal prion proteins by interacting with the two N-terminal domains of prion proteins, thereby reducing the incidence of prion infection [35].

## 6. The Application of Melanin

### 6.1. Applications in Agriculture and Industry

In industry, melanin can be used as an adsorbent for heavy metal pollution control. Heavy metals in various chemical forms and states are highly migratory, enriched, latent, and biotoxic and accumulate when they enter the environment and human body, threatening ecological environment safety and human health [194]. The prevention and control of heavy metal pollution are important to solve environmental problems. Compared with other adsorbents, nanoparticles have a higher reaction rate and adsorption capacity due to their small size, large specific surface area, high surface energy, and chemical activity. As a typical natural, non-toxic nanoparticle [195], melanin has a large specific surface area, and the structure surface has many functional groups and active sites that can bind to heavy metal ions. Secondly, melanin itself can be uniformly dispersed but insoluble in water, which facilitates subsequent operations such as solid-liquid separation. This excellent property of melanin can be effectively utilized and applied in various industries.

It has been proved that melanin has an ideal effect as an adsorbent, which can effectively remove Hg, Cr, Pb, Cu, and other heavy metals in a short period of time (3 h). In addition, melanin also shows good adsorption capacity when the concentration of heavy metals in the solution is lower than 10 mg/L. The adsorption capacity even surpasses some commercial adsorbents currently available on the market [185,196]. Melanin can also adsorb some organic drug molecules because of its diverse and heterogeneous structure. Melanin can bind widely used drugs in high concentrations, such as chloroquine, amlodipine, atorvastatin, and telmisartan. The binding amount is positively correlated with the drug concentration [197,198]. In agriculture, melanin can also be used as a pesticide photoprotective agent, prolonging the action of drugs because of the good light absorption effect. Melanin can reduce the influence of light and climate conditions on the biopesticides of Bacillus thuringiensis and is an ideal photo-protectant of biopesticides [199].

### 6.2. The Application in the Food Industry

Food packaging is an important part of the food industry. Its main role is to protect food from various external factors, such as temperature, ultraviolet light, humidity, oxygen, pressure, microorganisms, etc., in order to maintain the quality of food and extend its shelf life. Melanin has effective antioxidant, anti-radiation, and antibacterial activities, which can be used in food packaging to extend the shelf life of food products. The addition of melanin to the PLA films can improve the antioxidant and antibacterial activities and enhance mechanical strength and air-tightness [200]. The addition of melanin to PVA films can enhance the UV shielding and antioxidant properties [201]. When melanin isolated from squid ink is added to a nanocomposite film based on carrageenan, the thermal stability, UV radiation resistance, and the ability to resist foodborne pathogens are enhanced [202]. Low-density polyethylene films mixed with melanin can significantly enhance thermal stability and UV shielding, as well as water and oil resistance [203]. As discussed, melanin has a promising future in food packaging.

### 6.3. Biomedical Applications

Melanin can significantly increase the levels of specific and non-specific (including carbon scavenging) immunity, humoral immunity, and cellular immunity [29]. Calf RPE melanin can be involved in the regulation of retinal immune response, as evidenced by a significant increase in IL-6 secretion and expression in retinal epithelial cells [204]. Melanin extracted from the surface of Aspergillus fumigatus conidia combined with surfactant protein D exerts a PAMP effect, stimulates the secretion of proinflammatory cytokines, activates the host immune response, and promotes the spores to be more effectively phagocytic by macrophages [205]. This unique function could have promising applications for the modulatory treatment of immunocompromised patients and the development of new immune-enhancing nutraceuticals or drugs.

Melanin has the anti-tumor activity of promoting apoptosis and inhibiting angiogenesis. It can effectively improve the body weight, liver, spleen, and thymus index of mice with cancer and effectively regulate inflammation-induced high levels of various proteases, creatinine (CRE), blood urea nitrogen (BUN), and uric acid (UA) without causing toxic and side effects on other normal cells or tissues of the body [16]. Due to its good light absorption effect, melanin can absorb light from the ultraviolet region to the near-infrared region and has good photothermal conversion efficiency, which can be used as a photothermal agent for multimodal imaging-guided photothermal therapy (PTT) to selectively destroy tumor cells or tissues [40]. It has been reported that a transdermal microneedle patch containing melanin generates heat and promotes the uptake of tumor antigens by dendritic cells under near-infrared light. It also enhances the activity of anti-tumor vaccines and promotes anti-tumor immune responses [41].

Melanin or melanoidin themselves have good metal ion chelation properties and can bind various paramagnetic metal ions, such as gadolinium, iron, manganese, etc., without causing toxicity to humans. They have ideal biocompatibility and biosafety and are potential materials for T1 contrast agents in the MRI [36]. The longitudinal relaxation rate of MNP-Mn particles is significantly higher than that of Omniscan, which is about three times higher than that of gadolinium diamide. The intensity of the MRI signal reaches a maximum of 3 h after injection. Moreover, analysis with the cck-8 kit showed that MNP-Mn particles are virtually non-cytotoxic [39]. In addition to binding metal ions, melanin can also enhance the signals of MRI by binding liposomes. After intravenous injection of LIP-Mel in living mice for a certain time, the MRI images of the tumor site become bright, and the MRI signals are gradually enhanced [206].

### 6.4. Application in the Cosmetics Industry

The development and application of natural active ingredients are favored in the cosmetic industry. Melanin is a natural nanocomponent with a variety of active ingredients. Firstly, melanin, as a non-toxic natural pigment, can be used as a natural colorant in cosmetics. Secondly, the antioxidant and anti-radiation activity of melanin can be added to cosmetics as a photoprotective agent to prolong the aging time of cosmetics. In addition, it has a certain opsonizing effect on the skin, acting as an anti-aging agent, similar to vitamin C and vitamin E [159,207].

## 7. Conclusions and Outlook

Melanin, as a natural, non-toxic biological pigment from a wide range of sources, has attracted more and more attention from academia. In this paper, we systematically and comprehensively summarize the current status and importance of research on the classification and basis, sources and existence, degradation pathways, structure and characterization, biological activities, and applications of melanin. Melanin has a wide range of sources in nature. It exists in a variety of organisms, such as animals, plants, and microorganisms, and is easy to obtain. The indole structure and phenolic substances in the structure of melanin give it a variety of biological activities, such as antibacterial, antioxidant, free radical scavenging, chelating metal ions, anti-tumor activities, etc., which make it promising for applications in industries such as agriculture, food processing, cosmetics, and biomedical materials. However, there are still some issues to be explored before the formal large-scale application.

Considering the complex polymerization mechanism and process in the early stage of melanin synthesis, it is necessary to develop simpler, more economical, precise, and controllable large-scale acquisition procedures to obtain melanin with consistent structural properties to meet large-scale commercial needs. Then, the macromolecular structure of melanin is still unclear, and more innovative characterization techniques are needed to more precisely control the polymerization process of melanin and to further analyze its structure. For the biomedical applications of melanin in humans, its distribution in vivo, specific metabolic pathways, and potential long-term effects on organs and the immune system remain unclear, despite evidence of its good biocompatibility. More studies are needed to gain a deeper and clearer understanding of melanin and further determine its great potential for biomedical technology applications.

## Figures and Tables

**Figure 1 ijms-24-04360-f001:**
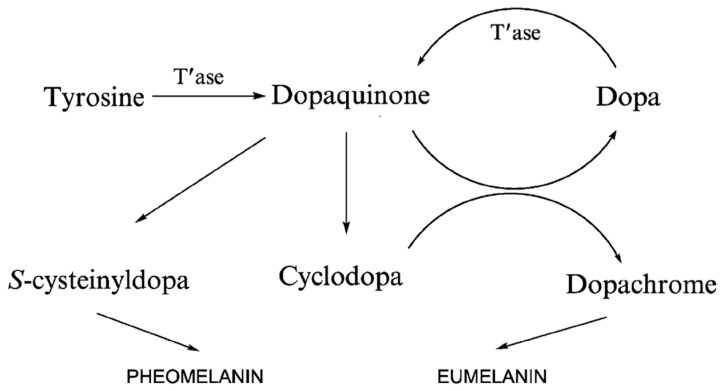
Divergence of melanogenic pathways. Adapted from [57] with minor modifications.

**Figure 2 ijms-24-04360-f002:**
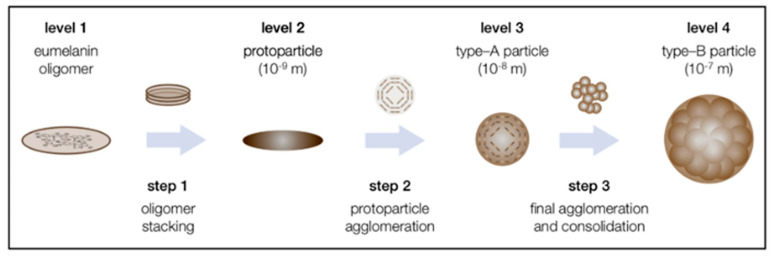
The proposed three-step mechanism for the supramolecular buildup of eumelanin based on the state of the literature and the presented results. Adapted from [56] with minor modifications.

**Figure 3 ijms-24-04360-f003:**
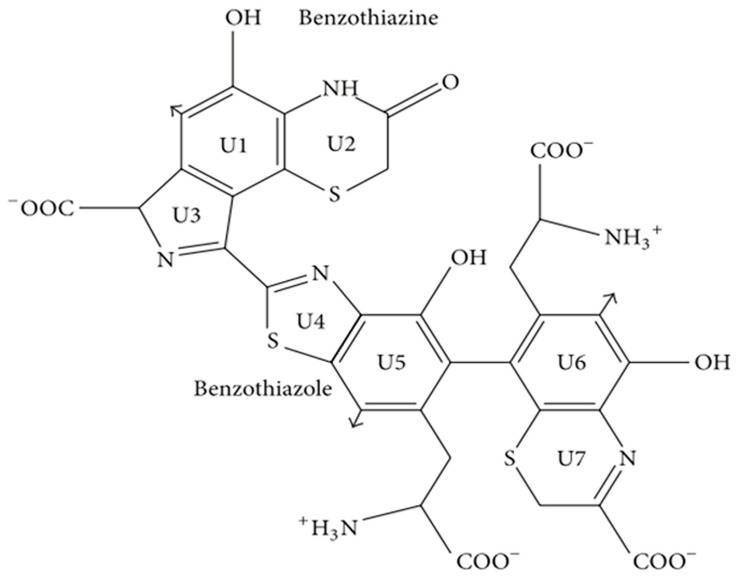
Structural diagram of benzothiazine and benzothiazole. Adapted from [3] with minor modifications.

**Figure 4 ijms-24-04360-f004:**
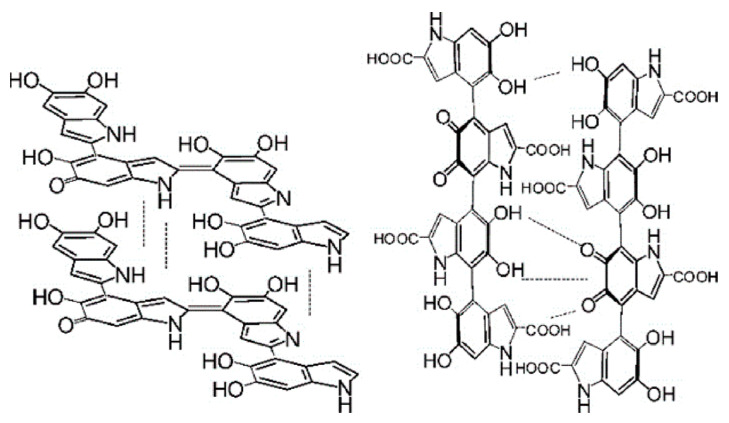
Representative structures of DHI and DHICA melanin. Left: DHI, right: DHICA. Adapted from [75] with minor modifications.

**Figure 5 ijms-24-04360-f005:**
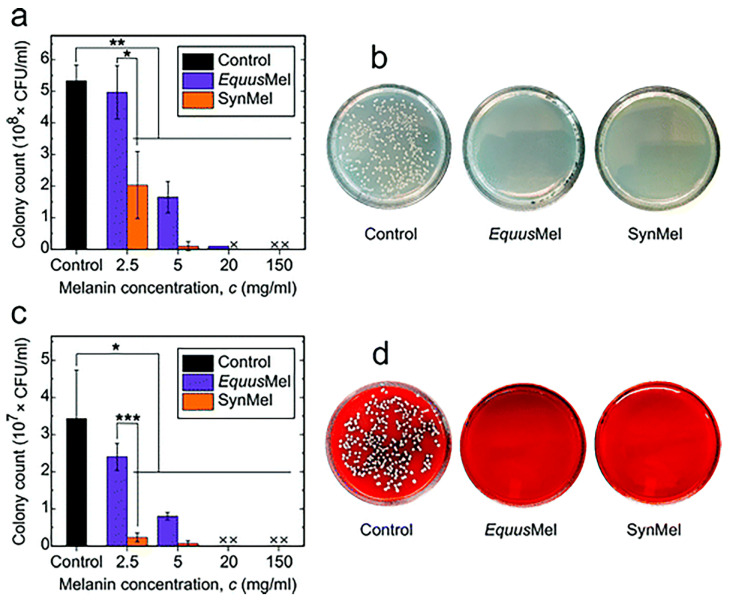
In vitro antibacterial activities are tested using (**a**) *E. coli* and (**c**) *S. aureus*. Statistically significant differences are indicated by *** *p* < 0.001, ** *p* < 0.01, and * *p* < 0.05 compared to the control. Representative images of agar plates are shown after incubating (**b**) *E. coli* and (**d**) *S. aureus* in melanin (150 mg mL^−1^) for 4 h at 37 °C. Reprinted with permission from Ref. [175]. Copyright 2021 Royal Society of Chemistry.

**Figure 6 ijms-24-04360-f006:**
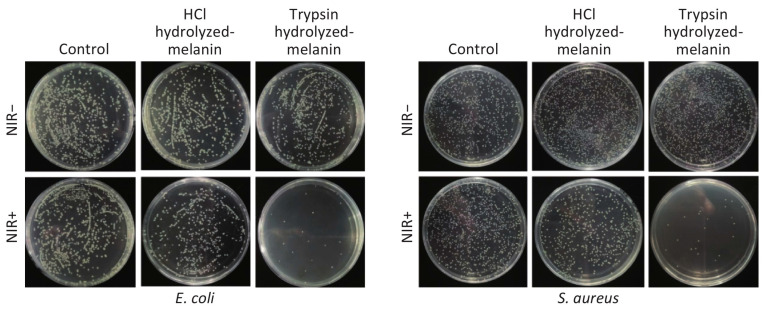
*Escherichia coli* (*E. coli*) and *Staphylococcus aureus* (*S. aureus*) were incubated with H-melanin and T-melanin, followed by NIR irradiation for 15 min. Photographs show *E. coli* and *S. aureus* after incubation with melanin, followed by NIR irradiation. Reprinted with permission from Ref. [177]. Copyright 2020 Elsevier.

**Figure 7 ijms-24-04360-f007:**
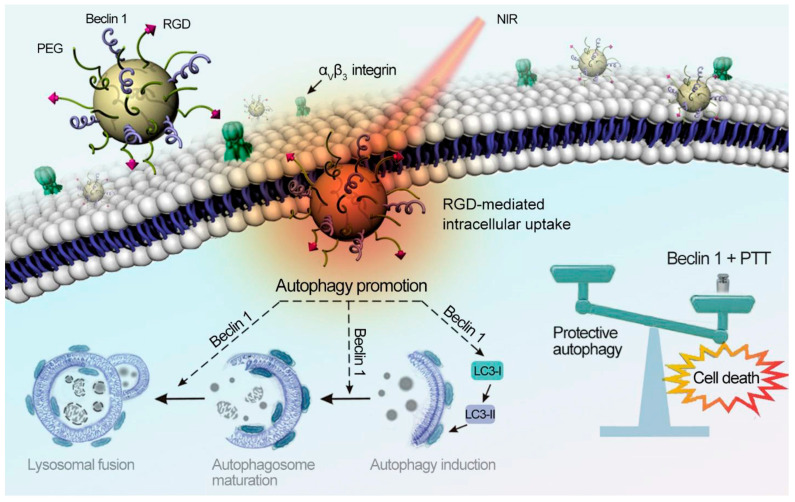
The illustration depicts 1-induced autophagy sensitizing photothermal killing of cancer cells. PPBR was internalized by cancer cells via RGD-αvβ3recognition. Beclin 1 on the nanoparticle surface up-regulated autophagy to sensitize cancer cells to PTT. Reprinted with permission from Ref. [34]. Copyright 2019 Elsevier.

**Table 1 ijms-24-04360-t001:** The five major melanins and their precursors. Adapted from [43] with minor modifications.

Pigment	Monomer Precursors
Eumelanin	tyrosine,5,6-dihydroxyindoles
Pheomelanin	cysteinyldopas, benzothiazines
Neuromelanin	dopamine, catecholamines
Allomelanin	1,8-dihydroxynaphthalene, phenolic precursors
Pyomelanin	homogentisic acid

**Table 2 ijms-24-04360-t002:** Isotherm parameters of heavy metal adsorption on Melanin. Adapted from [185] with minor modifications.

Heavy Metals	Langmuir Isotherm Model			Freundlich Isotherm Model		
	q_m_ (mg/g)	b (L/mg)	R^2^	K_F_ (mg/g)	1/n	R^2^
Hg	82.37	0.57	0.99	26.21	0.49	0.97
Pb	147.49	0.38	0.99	39.36	0.58	0.96
Cr	126.90	0.30	0.99	28.93	0.59	0.97
Cu	167.78	0.41	0.97	43.59	0.48	0.96

## Data Availability

No data available.

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
