# Peer review of "Recent Advances and Progress on Melanin: From Source to Application"

_ijms, 2023, doi:10.3390/ijms24054360_

Round 1
Reviewer 1 Report
This is an interesting review on melanin but please correct the mistakes whicu sometimes are serious.
Chapter 2.1 - please exchange for tautomer isomerase (separated words). Plese also mind that myxomycetes, though more of protozoans than fungi, also produce melanin in spores and even in the plasmodia (rarely).
Figure 1 (and further). Please mind that assumption that DOPA as a substrate for dopaquinone is an intermediate of L tyrosine (produced by tyrosinase) is a serious mistake. DOPA is produced further, non-enzymatically in the process of redox exchange. It may be produced by tyrosine hydroxylase to DOPA, it may be true, but NOT for tyrosinase, which produces directly DOPAquinone from tyrosine or DOPA (however, DOPA is not a product for tyrosinase). Please see and cite: Land et al. Met Enzymol 2004, DOI: 10.1016/S0076-6879(04)78005-2, Schallreuter et al. Exp Dermatol 2008, DOI 10.1111/j.1600-0625.2007.00675.x
Please correct the mechanism of melanin production accordingly through the text. Also in chapter 3.
Chapter 2.2 - please correct benzothiazide to benzothiazine through the text. Please also mimnd that the process of pheomelanogenesis may often turn in a reversed cyclization to benzothiazole, which places the free radical (if it is maintained) too far from heterocyclic nitrogen to generate hyperfine splitting of the EPR signal. Such melanins would be practically pheomelanins, though the signal looking like eumelanin. Many bird feathers contain benzothiazoles and reveal gray or dark coloration without yellow or red precence. It is therefore not precise to use the term"pheomelanin" to remark yellow or red coloration. It is more precize to call them benzothiazine- or benzothiazole-like coloration.
2.3 neuromelanin - sometimes the granules containing neuromelanin of pheo-and eu- components are called nigrasomes. Please use it.
5.1 Activity of Free radical scavenging
Please also cite papers by Matuszak and Reszka, who for the first time suggested that melanin may inhibit the activity of nitric oxide and its toxic metabolites. e.g. Matuszak et al. DOI 10.1515/nuka-2015-0084, Reszka et al., DOI 10.1016/S0891-5849(98)00058-6, and others.
5.2 Antimicrobial activities
please correct the words for: melanin, mussel shell (small case).
6 - The application of melanin
please do not use the term "very ideal" as it is the same as - ideal. The Ideal is always "very", - in the highest degree.
Reviewer 2 Report
This is a timely review that would have a potential to be of interest to the readers. However, it requires revisions in the structure and content.
Importantly, scientific English has to be corrected by an expert since some statement that sound erroneous could be secondary to deficiency in English. Also the authors should make an effort to cite more representative references.
The major recommendations are listed as follows:
In the historical overview of melanin, give credit to the funders. For example mention Raper Mason pathway with original citation. Mention seminal work of Lerner, who described tyrosine hydroxylase hydroxylase of tyrosinase. Mention seminal Pawelek's work on postdopa-oxidase steps of tyrosinase.
The major weakness is description of biochemistry of melanogenesis including description of catalytic activity of mammalian tyrosinase which includes hydroxylation of tyrosine, dopa oxidation and oxidation of dyhidroxyindole in first and last reactions dopa serves as the co-factor.
There is pH dependence, also metal cations can drive melanogenesis.
Role of melanin precirsrosrs such as L-DOPA and L-tyrosine is not adequately discussed. They can play various bioregulatory roles (Pigment Cell Melanoma Res 25, 14-27, 2012; J Theor Biol 164, 103-120, 1993).
The subcellular location of melanogenesis with description of melanosomes formation and their transport to keratinocytes is deficient. Also part on melanin degradation in epidermis is weak. The protein core of melanoproteins is degraded but not melanin itself.
Hormonal regulation of melanin synthesis requires adequate description and analysis (Physiol Rev 84, 1155-1228, 2004; Endocrine Rev 34:827-884, 2013).
The authors are encouraged to discuss melanin in the context of neuroendocrine properties of the epidermis (American Journal of Physiology-Cell Physiology 2022 323:6, C1757-C1776).
One of main functions of melanin is protection against solar radiation. Therefore, the readers will appreciate overview of various effects of UVR (Endocrinology 159(5), 1992-2007, 2018).
The authors emphasize the "tumor inhibiting activity of melanin". Unfortunately, this is a context dependent as described in reference 31, which is also good source to overview melanogenesis in general.
Note, that because of the intrinsic properties melanin can serve as negative factor in advanced melanomas (Oncotarget 20:17844-1785 2016 Feb 3. doi: 10.18632/oncotarget.7528; Int J Cancer 124, 1470-1477, 2009; Anticancer Res 18, 3709-3716, 1998) Hum Pathol 44 : 2071-4, 2013).
Melanogenesis can affect cellular metabolsim and local homeostasis (Arch Biochem Biophys 563:79-93, 2014; Exp Dermatol 24: 258-259, 2015) while intermediates have immunosuppressive effects.
There are also many sentences that difficult to understand, see below
"These two intermediates of dopa quinone and cyclodopaare rapidly redox dismutation to dopa and dopa pigment"
"Dopa is recruited to the pathway through the action of tyrosinase and dopa pigments continue to form the dark/brown eumelanin pathway"
There are typographical errors in the text, and in cited reference list.
Also number each line, for revision to make it easier to suggest corrections
Round 2
Reviewer 1 Report
The paper is much better and accepted for publications, however, please check carefully once again English. Please correct the bibliography, as the format of the citations is not uniform, the names are sometimes written in smallcase.
Please find that pheomelanin is detected also in Amphibians, see Wolnicka-Glubisz et al., Exp Dermatol 2012, 21(7)537-540, 10.1111/j.1600-0625.2012.01511.x
Reviewer 2 Report
Authors revised the manuscript appropriately.
I think this manuscript is acceptable in IJMS.
